behaviour, cognition, computational biology

social information use, consensus, polarization, socail learning, heuristics, cultural evolution

**Author for correspondence:**
Lucas Molleman
e-mail: l.s.molleman@uva.nl

†Equal contributions.

### PUBLISHING

# Strategies for integrating disparate social information

Lucas Molleman[1,2,3,†], Alan N. Tump[1,†], Andrea Gradassi[2], Stefan Herzog[1], Bertrand Jayles[1], Ralf H. J. M. Kurvers[1] and Wouter van den Bos[1,2,3]

[1]Center for Adaptive Rationality, Max Planck Institute for Human Development, Berlin, Germany
[2]Department of Psychology, and [3]Amsterdam Brain and Cognition Center, University of Amsterdam, Amsterdam, The Netherlands

LM, 0000-0001-7978-7173

Social information use is widespread in the animal kingdom, helping individuals rapidly acquire useful knowledge and adjust to novel circumstances. In humans, the highly interconnected world provides ample opportunities to benefit from social information but also requires navigating complex social environments with people holding disparate or conflicting views. It is, however, still largely unclear how people integrate information from multiple social sources that (dis)agree with them, and among each other. We address this issue in three steps. First, we present a judgement task in which participants could adjust their judgements after observing the judgements of three peers. We experimentally varied the distribution of this social information, systematically manipulating its variance (extent of agreement among peers) and its skewness (peer judgements clustering either near or far from the participant's judgement). As expected, higher variance among peers reduced their impact on behaviour. Importantly, observing a single peer confirming a participant's own judgement markedly decreased the influence of other—more distant—peers. Second, we develop a framework for modelling the cognitive processes underlying the integration of disparate social information, combining Bayesian updating with simple heuristics. Our model accurately accounts for observed adjustment strategies and reveals that people particularly heed social information that confirms personal judgements. Moreover, the model exposes strong inter-individual differences in strategy use. Third, using simulations, we explore the possible implications of the observed strategies for belief updating. These simulations show how confirmation-based weighting can hamper the influence of disparate social information, exacerbate filter bubble effects and deepen group polarization. Overall, our results clarify what aspects of the social environment are, and are not, conducive to changing people's minds.

## 1. Introduction

Social information guides decision making across a broad range of animal taxa [1–3]. By interacting with others and observing their behaviour, individuals can often glean useful cues helping them to learn the location of resources, acquire new skills, and adjust to novel circumstances [4–7]. The sources of social information available to individuals are largely determined by the structure of their social network [8,9]. How individuals gather and integrate information from their social environment shapes a number of key ecological and (cultural) evolutionary processes, including the transmission of knowledge through these social networks, the dynamics of social behaviour, and the emergence and persistence of local traditions [10–16]. In humans, social information use facilitates the accumulation of cultural knowledge across generations, which is widely deemed central to the ecological success of our species [17–19]. In recent decades, technological advances (most notably the Internet) have exponentially increased the number of potential sources of social information. While this affords instant

access to a wealth of useful knowledge, it is more likely than ever to encounter social sources holding disparate or conflicting views. In this paper, we examine the strategies that individuals use when they are confronted with such disparate social information coming from multiple sources.

In humans, social information use often involves changing one's mind after observing the behaviour of other individuals [20–23]. This process is commonly investigated using estimation tasks in which people are allowed to revise their initial estimates after observing the estimate of a peer (e.g. [24–28]). Studies using this approach give a detailed and quantified account of the effects of social cues on behaviour, primarily focusing on how individuals incorporate a single piece of social information [20–22,24,28,29]. Studies considering multiple peers have mainly described the effect of the central tendency (e.g. the mean of the pieces of social information provided [25,30–32]; but see [33]). In most real-world environments, however, people are confronted with multiple sources of social information at the same time, in various degrees of extremeness. Currently, it is unclear how people integrate such disparate social information. Here, we will address this issue in three steps.

First, we experimentally investigate how basic characteristics of the distribution of social information shape social information use. Specifically, we systematically manipulate the variance (reflecting the agreement among peers) and skewness (reflecting the clustering of peers close to or far away from the focal participant) of the distribution, while holding its mean constant. We show that the impact of social information strongly depends on its distribution. Disagreement among peers decreases its overall influence. Furthermore, the direction of the skew substantially alters the impact of social information: participants adjust their first estimate more when the majority of peers moderately agree with them and one peer strongly disagrees, compared to a situation in which a single peer strongly agrees with them, but the majority of peers strongly disagrees. This highlights the impact of confirmation-based weighting.

Second, we introduce a formal model to explain the strategies underlying these adjustments. This model is informed by previous research on individuals' strategies for incorporating a single piece of social information. This research has identified three distinct strategies: (i) keeping one's initial belief, (ii) adopting the behaviour of others, or (iii) 'compromising' between personal and social information [21–23,26,27,34–40]. We develop a modelling framework that extends these insights to situations with multiple social sources, accommodating both simple heuristics (keeping and adopting), and more complex strategies (compromising). Our model successfully recovers the main patterns in the observed data: it accurately predicts how the distribution of social information impacts the relative frequencies of adjustment strategies, and accounts for the strong between-individual heterogeneity in strategy use. Our modelling results reveal that social information receives more weight when it is in line with people's initial beliefs (reflecting confirmation effects), and when it is in close agreement with other social information (reflecting peer consensus).

Finally, we use our model to predict how the observed adjustment strategies may shape belief dynamics in exemplary social environments that vary in the like-mindedness of peers (e.g. due to the social network structure). These simulations reveal how and when people's prioritising of confirmatory social information can exacerbate filter bubble effects.

Moreover, they illustrate how individual differences in confirmation-based weighting can render beliefs to become more moderate (fostering group consensus) or more extreme (fuelling group polarization).

## 2. Experimental design

To examine how people integrate disparate information from multiple social sources, we used an adapted version of the BEAST (Berlin Estimate AdjuStment Task): a validated perceptual judgement task known to reliably measure individuals' social information use (figure 1) [29]. In the task, participants are shown several images of animal groups and have to estimate the number of animals (figure 1a,b). They then observe the estimates of three previous participants, and make a second estimate (figure 1c). The relative degree of adjustment quantifies an individual's social information use (figure 1d).

We study participants' social information use across four conditions that systematically differ in variance and skewness, while controlling for the mean deviation from a participant's first estimate (figure 1e): (i) low variance, not skewed (LN); (ii) high variance, not skewed (HN); (iii) high variance, with a cluster of two peers relatively far from the participant's first estimate (HF); and (iv) high variance, with a cluster of two peers relatively close to a participant's first estimate (HC). These conditions encompass a broad range of distributions individuals may encounter when sampling their social environment. Across all conditions, the three pieces of social information always point in the same—and correct—direction (i.e. avoiding situations in which the social information brackets the personal estimate). Importantly, holding constant the mean relative deviation from a participant's first estimate across conditions implies that a participant weighting all peer estimates equally should make similar adjustments across all conditions.

Prior to the main experiment, we pre-recorded individual estimates for each of the images by 100 individuals recruited from Amazon Mechanical Turk (MTurk), rewarding them for accuracy (electronic supplementary material, §3a). We used these estimates as social information in the main experiment. In a given round of the main experiment, the three pieces of pre-recorded social information were selected based on the participant's first estimate and the experimental condition of that round. That is, we selected those pieces of social information that most closely matched the experimental condition. This procedure allowed us to achieve experimental control without using deception (for full details and screenshots, see electronic supplementary material, §§3a and 4).

Ninety-five participants (all from the USA; 57% male; mean ± s.d. age: 35.8 ± 10.7 years) were recruited from MTurk for the main experiment, and completed 30 rounds of the judgement task. These 30 rounds included five rounds of each condition and 10 'filler' rounds. The social information in the filler rounds consisted of three randomly drawn estimates of the pre-recorded participants (for a given image). This procedure ensured that across all rounds, social information was sometimes higher and sometimes lower than a participant's first estimate, and sometimes bracketed the participant's own estimate. Reducing the regularity of the presented social information was expected to increase its trustworthiness. The 30 rounds were shown in a random order (and this order was the same for all participants). Throughout the task, participants did not receive feedback about their accuracy, impeding

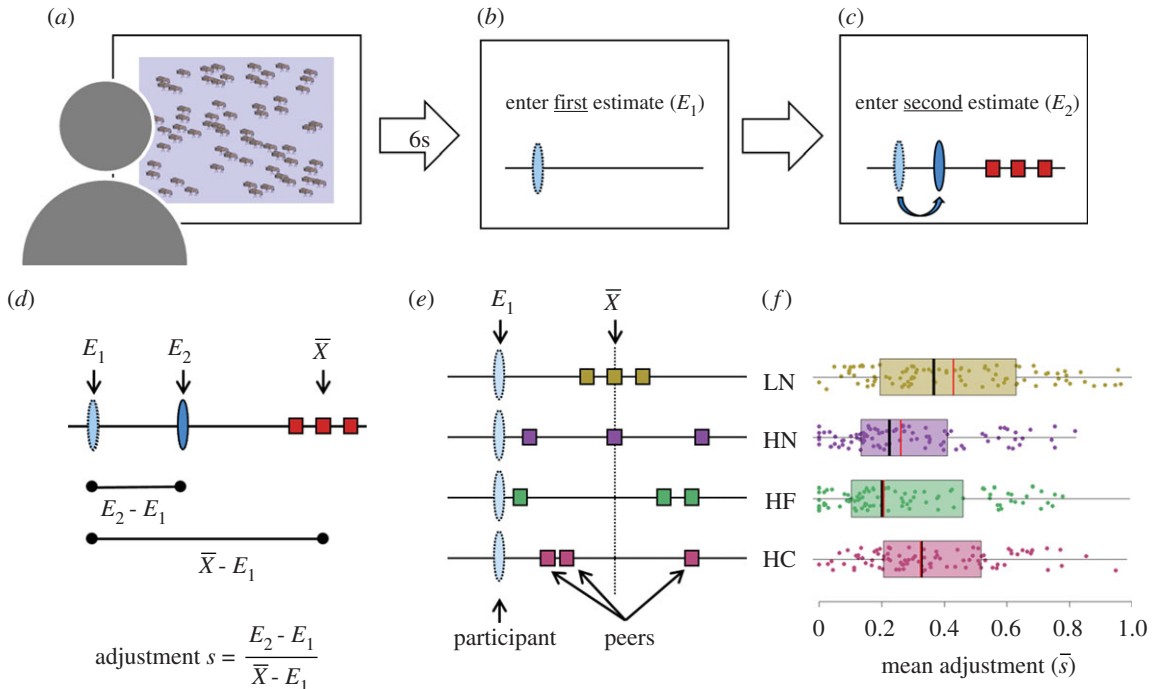

**Figure 1.** Experimental paradigm and the impact of disparate social information. (*a*) Participants start with observing a group of animals for six seconds. (*b*) Next, they enter their first estimate of the total number of animals using a slider. (*c*) Then, they observe the estimates of three pre-recorded peers (red squares), as well as their own first estimate (light blue oval), and enter their second estimate (dark blue oval). (*d*) Social information use in a round (*s*) is calculated as the adjustment from the first estimate ($E_1$) to the second estimate ($E_2$), divided by the distance between the first estimate and the mean of the social information ($\bar{X}$). Rearranging the terms highlights that $E_2$ is an average of $E_1$ and $\bar{X}$, weighted by *s*: $E_2 = (1 - s) \cdot E_1 + s \cdot \bar{X}$. (*e*) We varied the distribution of social information (squares) relative to a participant's first estimate (oval). Across four conditions, we manipulated the variance and skewness of the social information, while fixing the distance between the mean of the social information and the participant's first estimate (for details see Experimental design). Peer estimates displayed either low variance (LN) or high variance but no skewness (HN), or high variance with a skewed distribution, with a cluster of two peers far from (HF), or close to (HC) $E_1$. (*f*) Mean estimate shifts in each condition. Coloured dots show participants' mean adjustments across the five rounds of each condition; $\bar{s} = 1$ indicates a mean estimate shift to $\bar{X}$. Boxplots show the interquartile range (IQR), the median (black line) and the 1.5 IQR (whiskers). Red vertical lines show for each condition the predicted medians of the best-fitting model (see 'Cognitive model'). For strategies underlying mean estimate shifts across rounds, see figure 2. (Online version in colour.)

opportunities to learn about their own performance or the quality of the social information. Participants were rewarded for ccuracy: at the end of the experiment, one (first or second) estimate from the 30 rounds was selected and used for payment (see electronic supplementary material, §3a for details).

As a control, participants completed an additional block of five rounds (order counterbalanced) in which they did not observe the stimulus, but only the estimates of four peers. The distribution of these peer estimates emulated the distributions of the four experimental conditions (i.e. one of each condition), plus one filler round. This enabled us to compare how participants integrate four pieces of information of which none is their own, versus four pieces of information of which one is their own [41,42].

## 3. Results

### (a) Experimental results

Participants' use of social information strongly depended on its distribution (figure 1*f*). Participants adjusted their estimates most when social information had low variance and no skewness (LN condition; figure 1*f*, yellow), shifting, on average, 42% towards the mean social information. In the high variance and no skew condition (HN condition; figure 1*f*, purple), average adjustments were credibly lower (mean adjustment: 29%; see electronic supplementary material,

table S1 for statistics). Although adjustments in both conditions with skewed distributions were credibly lower than in the LN condition, the direction of the skew affected the relative adjustment: participants adjusted credibly more when two peers clustered relatively close to (HC condition; mean: 37%; figure 1*f*, red) rather than far from the participant is own estimate (HF condition; mean: 28%; figure 1*f*, green). Overall, these results demonstrate that variance and skewness in peer behaviour markedly affect social information use.

These results also show that average adjustments tended to be much smaller than what would be expected when participants weigh each of the three peer estimates as much as their own personal estimate (in which case they would adjust to the arithmetic mean of the four estimates, shifting 75% towards the mean social information in each condition). Moreover, participants substantially varied in their average adjustments, and these average adjustments strongly correlated across conditions (all pairwise Pearson correlations $r \geq 0.76$; electronic supplementary material, figure S1), indicating consistent inter-individual differences in social information use (see also below).

Figure 2 zooms in on the strategies underlying behavioural adjustments across rounds, differentiating between three distinct strategies: (1) *keeping* the first estimate, (2) *adopting* the estimate of one of the three peers or (3) *compromising* between the first estimate and the peer estimates. The relative frequency of these strategies differed markedly between the

Proc. R. Soc. B **287**: 20202413

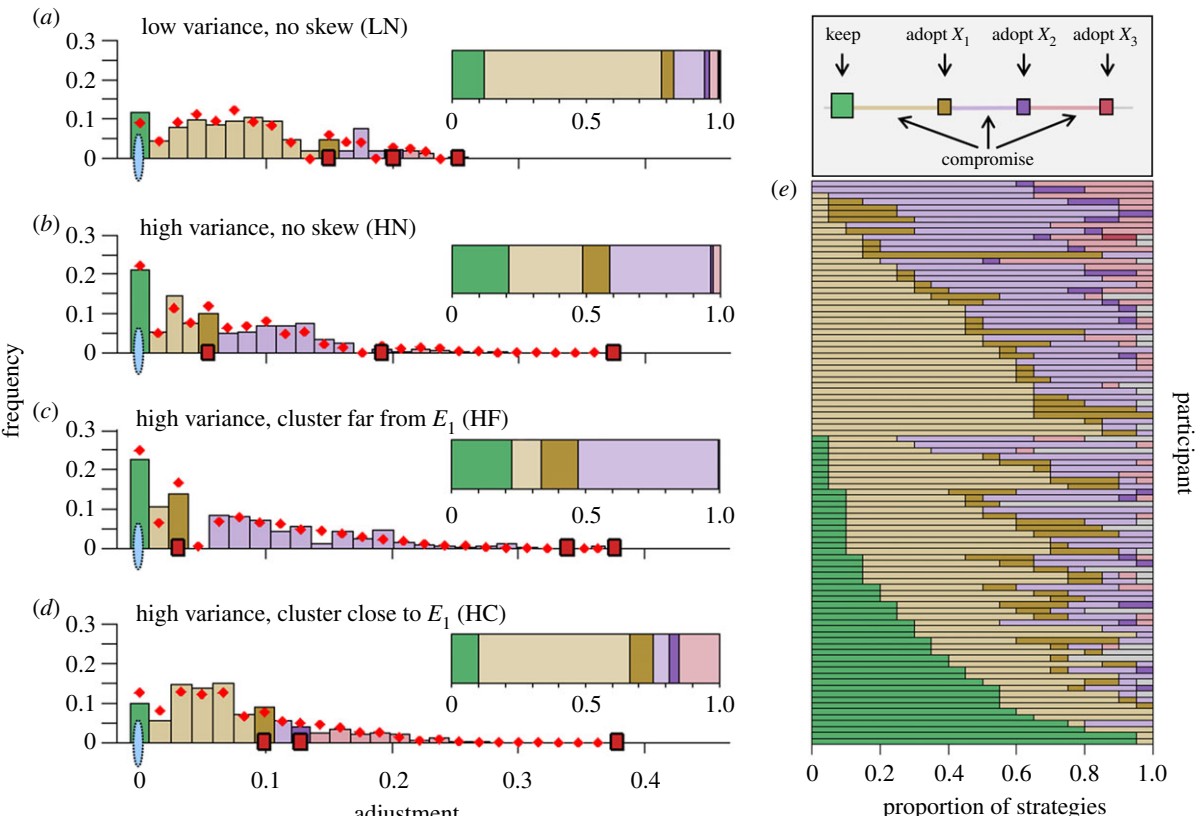

**Figure 2.** Adjustment strategies across conditions and participants. (a–d) Bars indicate the observed distribution of adjustments in individual rounds, expressed as the fraction of a participant's first estimate (i.e. $|E_2 - E_1| / E_1$), per condition. The relative positions of the peer estimates (red squares) slightly varied across rounds (shown are their mean positions; see Experimental design and electronic supplementary material, §3a for details). Bar and inset colours indicate the three strategies (i.e. keep, adopt and compromise). We observe that participants frequently kept their first estimate (green bars at $x = 0$), especially in the HN and HF condition. Across all conditions, compromising between personal and social information (light coloured bars) was the predominant strategy, whereas adopting the estimate of a peer (dark coloured bars at $x > 0$) was less common. Insets show the proportion of strategies per condition. Red diamonds show the predictions from the best cognitive model (see §3b), closely tracking the observed distributions. (e) The proportion of adjustment strategies for each participant (i.e. row) across all conditions (sorted according to their frequency of using the respective strategies shown in the legend: first by frequency of keep, then by frequency of compromise towards $X_1$, then by frequency of adopting $X_1$, etc.). (Online version in colour.)

four conditions (figure 2a–d; see electronic supplementary material, table S2 for statistics). When participants observed a single peer that closely agreed with them (HN and HF conditions; figure 2b,c), participants were more likely to either keep their first estimate, or to adopt the estimate of this near peer. When none of the peers was in close agreement with them (LN and HC conditions; figure 2a,d), participants were more likely to compromise, adjusting their estimate towards—but rarely beyond—the nearest peer. These results demonstrate that variance and skewness in peer behaviour have strong effects on the strategies people use to integrate social information. Figure 2e shows the frequency of strategies per participant across all conditions, illustrating that participants ranged from almost exclusively compromising, to exclusively keeping, with compromising being the most frequent strategy.

In all four control conditions—in which participants did not observe the stimulus, but four peer estimates, emulating the four distributions of the experimental conditions—responses were close to the arithmetic mean of the four peer estimates (electronic supplementary material, figures S2 and S3). Participants did, however, assign more weight to estimates closer to each other (electronic supplementary material, figure S4). This indicates that the observed deviations from the arithmetic mean in the experimental conditions (figures 1f and 2a–d)

are not due to an inability to integrate multiple pieces of information. Rather, the stark differences between the experimental and control conditions show that people down-weight social information that is more distant from their own first estimate, an effect known as 'egocentric discounting' [22,23,25,27,28,33,36–38,43].

## (b) Cognitive model

To investigate potential cognitive mechanisms underlying individuals' integration of disparate social information, we developed a set of models unifying simple heuristics (i.e. keeping and adopting) and more complex strategies (i.e. compromising; figure 3a). Based on our behavioural findings and previous literature, we assume that an individual selects an adjustment strategy (keep, adopt or compromise) depending on the distance between its own first estimate and the estimate of the nearest peer [33,43]. We further assume that, when compromising, individuals take a weighted average of their own first estimate and social information. We model compromising as a process of Bayesian updating (figure 3a). In this process, the weight of each of the peer estimates can depend on its distance to an individual's own first estimate (confirmation-based weighting [23,25,27,28]), and its distance to other peer estimates (proximity-based weighting; electronic

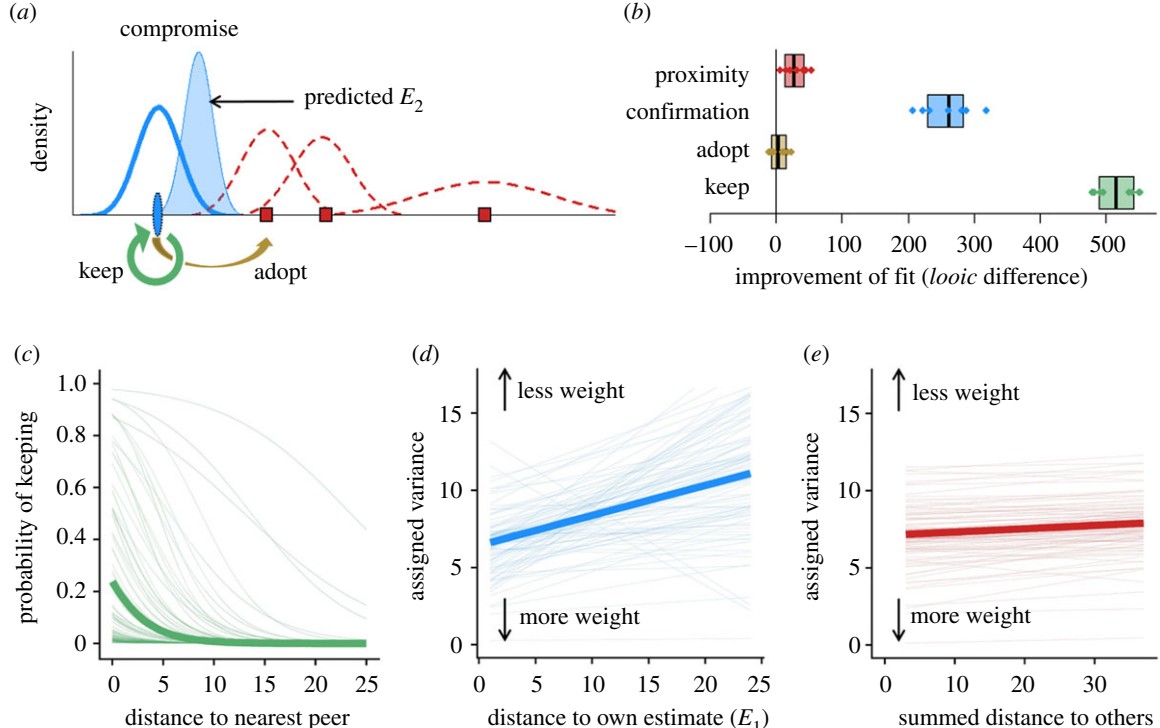

**Figure 3.** Cognitive model of the integration of disparate social information. (*a*) We model participants' estimate adjustments by combining heuristics of keeping and adopting with compromising strategies. We model the probability of keeping the first estimate (green arrow) as a function of the distance between a participant's first estimate ($E_1$; blue oval) and the nearest peer ($X_1$). Similarly, the probability of adopting the nearest peer estimate (brown arrow) is a function of this same distance ($X_1$ to $E_1$). Compromising entails taking a weighted average of $E_1$ and the peer estimates ($X_i$; red squares): we predict a participant's second estimate ($E_2$; transparent blue) based on Bayesian updating using weighted means of $E_1$ and each $X_i$. Personal and social information are represented as probability density functions with means at the observed estimates, and variances—indicating subjective uncertainty—following a normal distribution. The variance assigned to a piece of social information is inversely related to its weight in the updating process, and depends on its distance to $E_1$ (i.e. degree of agreement with the participant; 'confirmation') and its summed distance to other social information (i.e. degree of agreement with others; 'proximity'). (*b*) Improvement of fit for each of the model features based on *looic* differences (leave-one-out cross-validation information criterion; see electronic supplementary material, §3b). Dots show fit improvements for pairs of models excluding and including each feature, and the box plots show the median improvement and IQR. (*c–e*) The effect of each feature included in the best-fitting model. (*c*) Fitted probability of keeping one's first estimate as a function of its distance to the nearest peer. (*d,e*) Variance assigned to peer estimates as a function of their distance to $E_1$ (*d*), and as their mean distance to other peers (*e*). Thin lines represent estimates for individuals and thick lines show group-level means. (Online version in colour.)

supplementary material figure S4 [33,43]). These assumptions are reflected in four model features, capturing the selection of (i) the keep heuristic, or (ii) the adopt heuristic, and, when compromising, the weighting of social information based on (iii) confirmation or (iv) proximity. In electronic supplementary material, §3b, we provide full details of the model implementation and analysis.

We fit a series of models to simultaneously estimate the parameter values defining participants' selection of adjustment strategies (keeping, adopting or compromising), and confirmation- and proximity-based weighting. To account for individual differences in strategy use (figure 2*e*), we implement hierarchical models. We evaluate the importance of the four model features: the 'keep' and 'adopt' heuristics, as well as 'confirmation-' and 'proximity-based weighting', by calculating the leave-one-out cross-validation information criterion (*looic* [44]) of the 16 models comprising all possible combinations of these features (electronic supplementary material, table S3). A parameter recovery analysis confirmed that our model fitting procedure yielded robust and interpretable parameter values (electronic supplementary material figure S5 and §3b).

Figure 3*b* compares the fits of models including versus excluding each feature. It reveals that all features—except the adopt heuristic—reliably improve the model fit. Accordingly,

the best-fitting model includes the keep heuristic, and compromising with confirmation- and proximity-based weighting (electronic supplementary material, table S3). Figure 3*c–e* shows the effects of these three features in this best-fitting model (see electronic supplementary material, table S4 for the parameter estimates). Figure 3*c* shows that participants were more likely to apply a heuristic of 'keeping' when the nearest peer was in close agreement with them. Figure 3*d–e* illustrates the process of compromising, showing how confirmation and proximity impact the weight that is assigned to social information.

When compromising, participants tended to weigh personal information more than social information (electronic supplementary material, table S4), and participants assigned more weight to peers who more strongly agreed with them (figure 3*d*). This result is indicative of confirmation-based weighting (i.e. favouring information that affirms one's beliefs). In addition, participants assigned more weight to peers who showed more agreement with other peers (figure 3*e*). This 'proximity' effect was, however, weaker than the 'confirmation' effect (as indicated by the shallower slope in figure 3*e* than in figure 3*d*). For each of these features, the model detects substantial individual differences (indicated by the thin lines in figure 3*c–e*), thus capturing the high

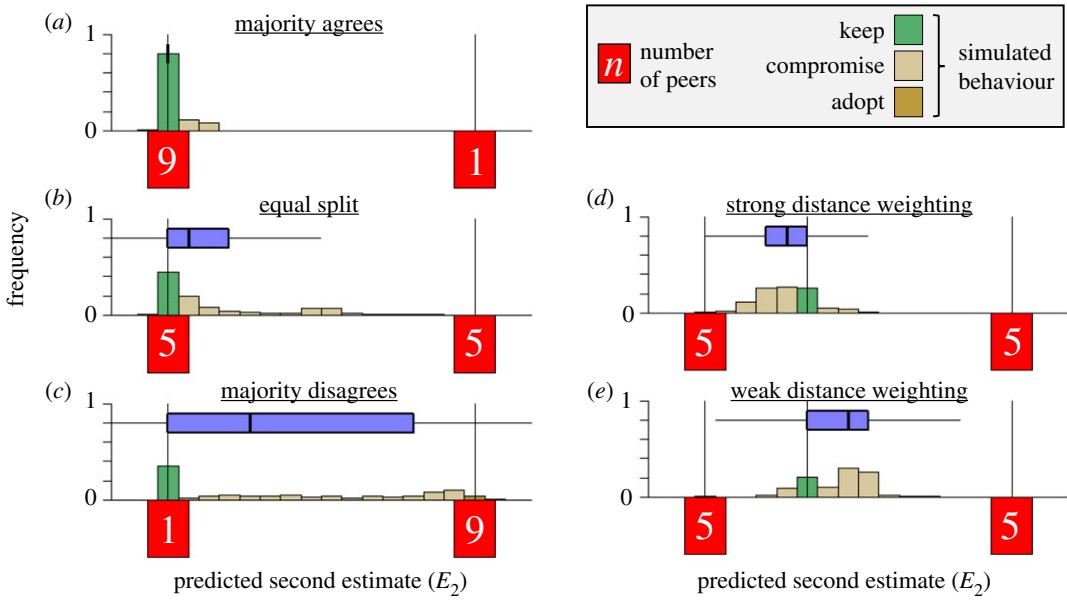

**Figure 4.** Simulated adjustments of agents observing ten pieces of social information in varying compositions. In the simulations, peer estimates were always low or high (vertical black lines; see electronic supplementary material, §3c for details). Numbers in the red boxes correspond to the number of peers in each category. In (a–c), the agent's first estimate is also Low. Agents likely keep their first estimate when they observe (a) a large majority agreeing with them, (b) half of the peers agreeing with them, and (c) only a small minority agreeing with them, though in the latter case agents shifted substantially more. In (d–e), peers are equally split, and the agent's first estimate is in between the two clusters of peers, but closer to one cluster than the other. (d) Agents with strong confirmation-based weighting (sampled from the upper 50% of the distribution; cf. steepest slopes in figure 3d) tend to adjust towards the local extreme estimate. (e) Agents with weak confirmation-based weighting (lower 50% of the distribution) tend to adjust towards the global mean estimate. In all panels, vertical bars show distributions of predicted adjustments across 1000 simulations. The horizontal boxplots summarize these distributions, showing the median, IQR and the 1.5 IQR (whiskers). (Online version in colour.)

inter-individual differences in mean adjustment and strategy use (figure 2f; electronic supplementary material, figure S1). Finally, we note that the absence of an effect of adopting the nearest peer estimate (figure 3b) may be because adopting can be mimicked by adjustment through compromising.

Importantly, the best-fitting model closely predicts the mean adjustment across conditions (figure 1f; red vertical lines) as well as the distributions of adjustments in rounds across conditions (figure 2a–d; red diamonds). This shows that the model can account for the main patterns observed in our experimental conditions. Our model also accurately predicts out-of-sample participants' mean adjustment and keep probability in the filler rounds (where peer estimates were randomly selected from the pre-recorded pool and frequently bracketed the participant's first estimate; electronic supplementary material, figure S6). Furthermore, the model can recover a commonly observed phenomenon in estimation tasks, namely that mean adjustments are highest when social information is at intermediate distance from first estimates (electronic supplementary material, figure S7 [25,27]). Taken together, these results suggest that our model can generalize to cases that are qualitatively different from our experimental conditions (on which the model was fitted).

## (c) Simulations

The identified strategies of social information use allow us to predict how they may shape belief shifts in settings where individuals encounter peers with various levels of like-mindedness. These settings can reflect individuals' access to information from their social network being local or global [45], their personal preferences for homophily

[46], or, in case of human online interactions, being controlled by algorithms prioritizing similar (or dissimilar) social sources over others, biasing the available social information [47]. In the following, we use simulations to explore how the social environment and social information use may foster consensus, or, alternatively, lead to polarization. We simulate agents who, as in the experiment, observe other estimates and adjust their first estimate. Agents observe 10 pieces of social information, across five qualitative different exemplary settings. We start with three settings in which an agent's first estimate is confirmed by either a (i) large majority, (ii) half of the peers, or (iii) only a small minority. We further simulate settings in which the focal agent is leaning towards one of two strongly disagreeing groups, and compare adjustment of agents with (iv) strong or (v) weak confirmation-based weighting. In each setting, we simulate 1000 agents whose adjustment strategies (i.e. their parameter setting) were sampled from the group-level distributions parameterized by the mean posterior estimates of the best-fitting model (see electronic supplementary material, §3c for details).

Figure 4 shows the predicted adjustments for the five scenarios. (i) When agents predominantly observe social information that agrees with their prior beliefs (as might happen in a 'filter bubble' [48,49]), they predominantly keep their first estimate or, at most, make very small adjustments (figure 4a). (ii) Even when only half of the peers agree with them—and the other half strongly disagrees (reflecting a typical attempt to 'de-bias' individuals [47])—agents only shift little, remaining far away from the global mean estimate (figure 4b). This suggests that even regular exposure to opposing information (e.g. from outside one's filter bubble) is unlikely to lead to substantial adjustments. (iii) Even when only a small minority

agrees with them, agents are still prone to keep their first estimate, although adjustments towards the majority become more substantial (figure 4c). Overall, these simulation results illustrate how confirmation-based weighting can curb belief updating: confirmatory social information reinforces people's prior beliefs and prompts them to retain these beliefs, even if they reflect minority views.

To further examine the possible implications of individual differences in confirmation-based weighting, we simulated adjustments for agents showing weak and strong confirmation effects (i.e. individuals with steep and shallow slopes in figure 3d). Agents were initially located in between two clusters of peers, but slightly closer to one of the clusters. In this setting, agents with strong confirmation-based weighting tend to adjust towards the local cluster—moving away from the global mean (figure 4d). Weak confirmation-based weighting tends to lead to adjustments towards the global mean estimate (figure 4e). These results suggest that strong confirmation-based weighting can drive people to more extreme views, and may increase polarization over time.

## 4. Discussion

This paper makes three novel contributions. First, we experimentally show that the impact of multiple sources of social information strongly depends on its distribution. Increased variance in social information reduces participants' adjustments, and skewness decreases adjustments if a single peer confirms their first estimate. Second, our cognitive model provides a unified account for how people integrate disparate social information, showing that people rely on a combination of simple heuristics of keeping their initial beliefs and compromising towards social information. The model captures how the weight of social information is determined by both its degree of confirmation of people's initial beliefs, and its proximity to other pieces of social information. Third, the model made accurate out-of-sample predictions of adjustments in qualitatively different judgement situations, and our simulations illustrate how prioritising confirmatory social information may lead individuals to take up more extreme beliefs.

Overall, people assigned more weight to their personal initial beliefs than to social information (figures 1f and 2a–d; electronic supplementary material, figure S3, figure S4 and table S4). It may seem somewhat puzzling why they would do so in a task in which social information consists of judgements of people incentivised to accurately solve the same problem. Indeed, there is no reason to assume that one's first estimate would be more accurate than those of others. One rationale for prioritising personal estimates is the lack of access to others' reasons for holding their beliefs, which may lead participants to discount social information, especially when it is very distinct from one's personal beliefs [28,50] (but see [41]).

Our cognitive model provides a detailed account of people's social information use, contributing to understanding its underlying computational and cognitive mechanisms [51]. The model presents a unified framework laying out how people integrate personal and social information, by combining a heuristic strategy (keeping) with compromising (weighted averaging). We further obtain a detailed picture of the process of compromising by formalizing how people weigh several pieces of

social information, the combined effects of which would be hard to understand without a model. Interestingly, while social information that is consistent with own personal beliefs is weighted more when people compromise (confirmation-based weighting; figure 3d), it also increases the chance that people simply keep their initial beliefs (figure 3c). The combined result of these effects is that social information tends to have the strongest impact on overall adjustments when it is at intermediate distance (electronic supplementary material, figure S7). Our results reveal that people also weigh social information based on its consistency with other social information ('proximity-based weighting'; figure 3e; electronic supplementary material, figure S4). One rationale for prioritising social information consistent with other social information is relatively straightforward: when people are motivated and able to make valid judgements, agreement among peers reliably signals accuracy [52,53]. The concerted action of the three mechanisms (heuristics of keeping, and compromising based on confirmation and peer proximity) explain our experimental observations that the impact of social information strongly depends on the variance and skewness of its distribution.

Our model's accurate predictions of behaviour in the filler rounds underscores its ability to go beyond mere redescription of the data it was fitted to, and suggests that our model can be generalized to settings that are qualitatively different from our experimental conditions (electronic supplementary material, figure S5 and figure S6). Our simulations go beyond the limited set of distributions of social information studied in our experiments, generating predictions about how social information use may be shaped by distributions resembling important real-world settings. The advent of the Internet has dramatically changed the structure and dynamics of social interactions; at times being (algorithmically) biased towards like-minded sources [47,49,54], but also giving people access to diverse social sources with potentially conflicting views. Our simulations predict that disparate social information changes people's minds only to a limited degree, even when this social information signals that people hold minority views (figure 4). More importantly, under certain conditions, observing balanced social information can even lead individuals with strong confirmation-based weighting to take more extreme views (figure 4d). These findings have direct implications for interventions. For instance, they suggest that efforts to debias online information that present people with balanced views [47] might not suffice to break filter bubble effects and dynamics of polarization. Future work could more explicitly address the role of the social network structure, testing how the distribution of beliefs across social networks may interact with network structure in governing the dynamics of belief updating processes in a population (e.g. [55]). We believe our cognitive modelling framework can help in achieving a mechanistic understanding of how social information use may contribute to the formation of group consensus or the risk of polarization, and more generally, how the distribution of individual strategies of social information use in a population drives the transmission of information across social networks and shapes the course of cultural evolution.

The current study provides a robust template for understanding social information use in a range of (complex) social environments that people encounter in their day-to-day lives. Future empirical work should test the predictions of our simulations, as well as the extent to which our findings—obtained with a stylised perceptual judgement

task with anonymous peers—generalize to other domains of decision making. Our results demonstrate that confirmation-based weighting can strongly reduce people's willingness to change their minds even in the minimal setting of a task with an objectively correct solution. It seems plausible that confirmation-based weighting—and its polarizing consequences—are even stronger in many important real-world contexts involving emotive, moral or political issues. In these (often controversial) contexts, the integration of disparate social information may be further hampered due to 'motivated reasoning' [56] or when observed individuals belong to an out-group [57,58]. Conversely, disparate social information might impact behaviour more strongly when it stems from peers who are familiar [59], similar [11,60], prestigious [61,62], or known to have expertise in the task at hand [7,63,64]. Moreover, the effects of each of these factors are likely to substantially differ between individuals (cf. figures 2*e* and 3*c*–*e*) and between societies [65–68]. Our experimental design and modelling framework are flexible enough to include adaptations to accommodate each of these elements, helping understand the features of social sources that may influence weighting. For example, if social sources vary in their expertise, individuals might assign more weight to social information provided by an expert rather than by a non-expert. By explicitly accounting for the amount of variance people assign to social information from experts versus non-experts, an extended version of our model could distinguish how the impact of social information depends on the source's expertise, beyond its degree of alignment with people's initial beliefs. Furthermore, our hierarchical modelling approach allows accounting for individual differences in social information use, a regularly observed but poorly understood phenomenon in humans and other animals [65,69–73]. Linking individuals' strategies and their underlying cognitive mechanisms to genetic, developmental and cultural processes may help unearth the causes of between-individual and between-society differences in social information use.

To conclude, our findings contribute to a growing literature on how people integrate social information to update their beliefs. We go beyond previous experimental work on the effects of social information (which focused on single social cues or the mean of multiple cues) by showing that the variance and skewness of its distribution strongly modulates its impact on behaviour. Our cognitive model provides a detailed picture of the cognitive mechanisms that underlie the integration of disparate social information, highlighting the role of heuristics of keeping one's initial beliefs, and the importance of confirmation- and proximity-based weighting. Finally, our simulations consider various exemplary social environments to illustrate how confirmation-based weighting can markedly exacerbate filter bubble dynamics and polarization. We anticipate that these findings will provide a useful point of departure for future work aiming to understand the nature of human social information use and its implications for group dynamics, and to inform interventions to effectively de-bias individuals and help them forming accurate beliefs about the world.

**Ethics.** Ethical approval was obtained from the Institutional Review Board of the Max Planck Institute for Human Development Berlin (ARC 2017/18).

**Data accessibility.** All code associated with this paper (i.e. experimental software, cognitive model, simulations and analyses) is publicly available in the public OSF repository https://osf.io/rmcuy/.

**Authors' contributions.** L.M., A.G., R.H.J.M.K. and W.v.d.B. designed the study. L.M. and A.G. programmed the experiment, A.G. collected the data, L.M., A.N.T., S.H. and W.v.d.B. developed and analysed the model. All authors were involved in interpreting the results, and writing and editing the manuscript.

**Competing interests.** We declare we have no competing interests.

**Funding.** L.M. and W.v.d.B. are supported by an Open Research Area grant (ID 176). L.M. is further supported by an Amsterdam Brain and Cognition Project grant 2018. W.v.d.B. is further supported by the Jacobs Foundation European Research Council grant no. (ERC-2018-StG-803338) and the Netherlands Organization for Scientific Research grant no. (NWO-VIDI 016.Vidi.185.068).

**Acknowledgements.** We thank Casper Hesp and the members of the Connected Minds Lab at the University of Amsterdam for useful comments and discussions.

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
