## [Reviewer comments · Proceedings of the Royal Society B: Biological Sciences]

Review History

RSPB-2020-1310.R0 (Original submission)

Review form: Reviewer 1

Recommendation

Accept with minor revision (please list in comments)

Scientific importance: Is the manuscript an original and important contribution to its field?

Excellent

General interest: Is the paper of sufficient general interest?

Excellent

Quality of the paper: Is the overall quality of the paper suitable?

Good

Is the length of the paper justified?

Yes

Should the paper be seen by a specialist statistical reviewer?

No

Do you have any concerns about statistical analyses in this paper? If so, please specify them explicitly in your report.

No

It is a condition of publication that authors make their supporting data, code and materials available - either as supplementary material or hosted in an external repository. Please rate, if applicable, the supporting data on the following criteria.

Is it accessible?

Yes

Is it clear?

Yes

Is it adequate?

Yes

Do you have any ethical concerns with this paper?

No

Comments to the Author

This concise, well-constructed study uses a powerful combination of experimental and theoretical approaches to study how the distribution of social information from multiple sources affects individual judgements. The topic is important and very timely given the ever-increasing importance of social media in influencing individual attitudes and behaviour, including in areas of vital importance for public health such as vaccine acceptance, mask-wearing, and so on. The statistical approach is sound. The paper is also extremely clearly written and presented, with excellent use of informative figures.

I would recommend only a small number of minor changes before the study is accepted for publication in Proceedings B.

Ecological validity: the experimental design is sophisticated and in several ways far better reflects social influence in natural conditions compared with similar predecessors. In particular, allowing participants to view the judgements of multiple 'demonstrators' and integrate social with personal information to form 'compromise' decisions is much more realistic than other designs common in the field such as 'either/or' tasks where participants must choose between individual judgements and social information (often from a single demonstrator). However, as for any experimental task, the authors should consider the ways in which the study does not well represent 'real-world' conditions. Here, the task of guessing the number of animals in a herd is quite far abstracted from the social and political contexts the authors wish to generalise their findings to. Most importantly, in this study's online task the demonstrators are anonymous strangers with no social connection to the participants, which may partly help to explain the relatively minor influence of social information relative to individual decisions the authors find here. While many contemporary online social spaces do involve interactions between anonymous strangers, individuals typically still have a consistent identity over time along with signifiers of expertise and affiliation which affect their level of influence. For example, my opinions on the state of the UK's legal system would probably be influenced far more by the "Secret Barrister" than by any completely random, anonymous stranger posting on Twitter. The authors should also consider potential effects of the presentation of the task over the findings. In particular, the presentation of social information as numeric quantities on a linear, sliding scale (ESM) has obvious advantages in terms of the clarity of the task to participants, but again does not well represent social influence in the real world, which often involves multiple, both qualitative and quantitative dimensions. For example, this task does not well represent the nature of social influence over somebody deciding which way to vote in an upcoming election. Of course, no experiment can perfectly represent real-world conditions, but the authors should give these

issues some consideration when evaluating their findings.

Representativeness: the main text includes little information about the study sample. The information on recruitment and participant demographics should be moved from the ESM to the main text, or at least briefly summarised there. Additionally, the authors should include some further detail on the ethnic/cultural make-up of the sample and provide descriptive statistics, if possible. This is important because the unrepresentativeness of 'WEIRD' samples is by now well-established (Henrich et al. 2010) and the use of social information has been shown to vary between cultures (e.g. Mesoudi et al. 2014).

Deception: the author's experimental design cleverly avoids the need for deception while maintaining control over the social information. This should be mentioned in the main text as well as the ESM as it is a strength of the study. The authors should, however, also consider the possibility that participants may not believe the social information is real, despite their best efforts, considering that deception is still quite common in online experiments.

Understanding checks: as presented (if I understand correctly!) it looks as if the correct answers for the understanding checks (ESM) are all selected by default. Is that right? If so, I'd suggest avoiding this in future as it allows participants to pass the check even if they are just "clicking through" without properly reading the questions.

Henrich, J., Heine, S. J., & Norenzayan, A. (2010). The weirdest people in the world?. *The Behavioral and brain sciences*, 33(2-3), 61-135.

Mesoudi, A. et al. (2014) 'Higher frequency of social learning in China than in the West shows cultural variation in the dynamics of cultural evolution', *Proc. R. Soc. B* 282: 20142209.

Review form: Reviewer 2

Recommendation

Major revision is needed (please make suggestions in comments)

Scientific importance: Is the manuscript an original and important contribution to its field?

Excellent

General interest: Is the paper of sufficient general interest?

Good

Quality of the paper: Is the overall quality of the paper suitable?

Acceptable

Is the length of the paper justified?

Yes

Should the paper be seen by a specialist statistical reviewer?

No

Do you have any concerns about statistical analyses in this paper? If so, please specify them explicitly in your report.

Yes

It is a condition of publication that authors make their supporting data, code and materials available - either as supplementary material or hosted in an external repository. Please rate, if applicable, the supporting data on the following criteria.

Is it accessible?

Yes

Is it clear?

Yes

Is it adequate?

Yes

Do you have any ethical concerns with this paper?

No

Comments to the Author

Here, I summarize my thoughts on the submission, "Strategies for integrating disparate social information" to Proc B by Molleman et al. The key idea behind this study is straightforward, compelling, and insightful. At the root of every cultural evolutionary process is a distribution over social learning strategies. For many years, researchers working in this field were modellers, and what we had was a profusion of models, none of them with an especially sound empirical basis. Luckily, for roughly 10 years now, we've had a growing tradition of controlled experiments specifically designed to identify the social learning strategies that drive cultural evolutionary dynamics. All in all, this new field is a welcome change, but it's also highly unsettled. There are many paradigms, with little to help us sort out the relative benefits and liabilities of the different paradigms. Into this mix, we have the present study, which takes a clever new approach to a fundamental problem. Basically, the decision-making task can be viewed as a task with a large but finite number of options. Specifically, subjects had to guess the number of animals presented in an image. The key innovation is this. After making an initial choice, subjects were presented with social information as a set of three choices by others. The subject could then revise. What's clever here is that the authors manipulated the distribution of choices among the three observed individuals. Fig 1e shows the basic manipulations. Intuitively, (a) three others, close to each other, (b) three others, far apart from each other. (c) three others, with one close to the subject's initial choice and two far away, and (d) three others, with two close to the subject's initial choice and the other far away. For all four treatments, the difference between the subject's initial choice and the average of the three observed choices was constant. The question then is, how do the properties of the distribution characterizing observed social information shape how people learn socially? This is an important question as it goes straight to core issues in the study of cultural evolution and gene-culture coevolution. Namely, how do we actually learn from others? In spite of what the modellers would say, we don't actually know much about this. In this study, the particular question is, how do we actually learn from others when we observe a distribution over a large option space (probably a common situation)? In addition, how do the ways in which we learn vary by individual and/or context? We really don't know much about this, although Mesoudi (2016) is a nice review, and the first author of this submission has also done some nice work on the topic.

The current study has considerable potential, and I encourage the authors to continue with the interesting and important line of research. But, the paper would require a number of careful revisions before being suitable for publication in Proc B. I detail below, with points more or less in the order they come up in the paper itself.

- Where's the biology? At present, the paper is written almost as if it's intended for a psychology journal, perhaps a sociology journal, or maybe a general science journal. In some ways, Proc B has the flavour of a general science journal. But finally, it's a biology journal. The biology is basically missing from the submission as currently formulated. Ordinarily, one might think that

this would be devastating for the submission, but I don't think that's true here. It would be easy - even trivial - to motivate the present study with fundamental questions related to evolutionary ecology, behavioural ecology, and in particular the evolutionary study of human social behaviour. For some reason, the authors did not do so for their initial submission, but I'm sure they can, and I believe they should. I would recommend discussing, at minimum, social learning and cultural evolution, heterogeneity in social learning, and the ever-controversial hypothesized link between culture and population structure.

- The description of the experiment in the main paper is missing key details. In particular, I couldn't tell from the main paper if choices/accuracy were incentivised, and I couldn't see how the authors could possibly implement the key treatment manipulations (LN vs HN vs HF vs HC) without using deception. As it turns out, choices were incentivised, and the authors did not use deception. I had to dig through the supplement to find this information, though, and even then the means by which the authors avoided deception still wasn't entirely clear. Regardless, given that incentivised choices and deception are two points of disagreement in contemporary experimental work, the authors should explain these details in the main paper instead of leaving the reader to wonder and speculate.

- On related notes about the experimental design, I could not tell from the main paper what kind of feedback participants had at the end of each round. From the supplement, I think, but I'm not entirely sure, that participants did not get feedback about how close their respective guesses in a given round were to the correct answer. If correct, this information should be included in the main paper. In effect, even if each round gets a new stimulus (i.e. image with some number of animals), feedback at the end of each round would still allow participants to learn which social learning strategies are relatively effective, which in turn could mean that social learning strategies evolve endogenously throughout the course of a session. If participants did not get this kind of feedback, then this kind of learning is not possible. Depending on one's objectives, neither design is clearly better than the other, but the reader needs to know which one was implemented. Finally, even without deception, the design, though clever, raises the following potential concern. In 20 out of 30 rounds, participants were provided with social information that followed extremely regular patterns. In effect, social information in these 20 rounds exhibited four highly regular and highly stylized distributions. I didn't get a strong sense how this was experienced by participants. The concern is that, if the social information looks too regular, participants might question whether it's real even if it is. The point is that the authors simply need to put more description of the experiment in the main paper. I know space is tight, so they'll have to be disciplined. But, experimentally minded readers will want a clear sense of the paradigm from a participant's perspective.

- As two final notes on the experiment/data, everything has been uploaded to OSF. However, from looking around on the repository, it looks like the authors did not pre-register the study. Rather, they simply uploaded everything over the course of two days after they were done. Also, they ran the study on mTurk. If I've understood correctly, neither of these choices is entirely consistent with current methodological trends related to experimental studies on evolution and human behaviour. At least, the authors should explain their reasons.

- Fig. 2 requires more explanation to be interpretable.

- The explanation of the statistical methods was too terse and too ambiguous to understand. The supplement didn't help that much. If I had to trace my difficulties to something specific, I would say that the challenge follows from a tendency to use jargon as if it's standard terminology understood by all. From p. 10 onward, I see two different paths, either of which would probably be an improvement. The first possible path would involve considerably fewer technical details in the main paper. Just give the basic idea, refer to the supplement (which should definitely have a much more explicit, complete, and detailed exposition of the technicalities), and move on to the results.

The second path would involve more technical details in the main paper. To illustrate, what exactly do the authors mean by a "Bayesian mixture model"? What is the "standard logistic function"? Especially confusing, the use of the phrase "standard logistic function" implies a function that generates a probability distribution over a support with cardinality two. The authors, however, are modelling an outcome space with cardinality three ($\{\text{keep}, \text{adopt}, \text{compromise}\}$). This led me to ask, why are they using logistic when multinomial would be more appropriate? The paper outlines two models, with $P(\text{keep})$ and $P(\text{adopt})$ respectively on the left-hand sides. Each of these models is based on the standard logistic function. I naively take this to mean that they specify two models, one of which models a probability distribution over $\{P(\text{keep}), P(\text{not keep})\}$, and the other of which models a probability distribution over $\{P(\text{adopt}), P(\text{not adopt})\}$. If correct, how does one ensure that $P(\text{keep}) + P(\text{adopt}) \leq 1$? I couldn't really form an opinion, however, because the key details weren't presented. Also, why are beliefs treated as discretised normal distributions? Why not simply use a distribution with non-negative integers as a support? Maybe this is something people do when they fit these kinds of models. I don't know. But the motivation for such an approach is not obvious. A normal distribution, of course, puts mass on negative numbers, which are obviously not possible. The need to discretise also suggests that normal is the wrong distribution anyway.

The issues mentioned here are representative of the basic challenge. The paper in its current form provides just enough technical detail to confuse, but not enough to ultimately sort it out and form an opinion in one's head. So, I would recommend one of the two paths outlined above. Regardless of which path, the full details should be in the supplement to a much greater extent than what we have now.

- The bottom of p. 11 is not a sentence. I didn't understand the intended meaning.

- Finally, the simulations also require more explanation. Also, this section offers an opportunity to make stronger ties to biology. As currently framed, the discussion of the simulations reads like something targeted at social psychologists or even political scientists. While we're all no doubt fascinated and perhaps disgusted (alternatively delighted) by the social and political fragmentation destroying certain powerful countries, I'm not sure this should trump a discussion about fundamental biological questions in a submission to Proc B. Much of the discussion here, for example, could be linked to basic hypotheses about culture, population structure, and the potentially surprising evolutionary dynamics that might follow.

Decision letter (RSPB-2020-1310.R0)

28-Jul-2020

Dear Dr Molleman:

I am writing to inform you that your manuscript RSPB-2020-1310 entitled "Strategies for integrating disparate social information" has, in its current form, been rejected for publication in Proceedings B.

This action has been taken on the advice of referees, who have recommended that substantial revisions are necessary. With this in mind we would be happy to consider a resubmission, provided the comments of the referees are fully addressed. However please note that this is not a provisional acceptance.

The resubmission will be treated as a new manuscript. However, we will approach the same reviewers if they are available and it is deemed appropriate to do so by the Editor. Please note that resubmissions must be submitted within six months of the date of this email. In exceptional

circumstances, extensions may be possible if agreed with the Editorial Office. Manuscripts submitted after this date will be automatically rejected.

Sincerely,
Dr Robert Barton
mailto:proceedingsb@royalsociety.org

Associate Editor
Board Member: 1
Comments to Author:

The two reviewers agree, as do I, that this study takes a very clever approach to address interesting and important questions in cultural evolution. However, both reviewers raise a number of important concerns that need to be addressed in revision.

In particular, both reviewers highlight a lack of detail on the likely ecological relevance of the study (as reviewer 2 puts it, "where's the biology"). Emphasising how the study may reflect real-life problems and linking it more clearly to the literature on social learning, cultural evolution and population structure will help to address these concerns.

Both reviewers also request further details and clarification on aspects of the experimental and analytical methods. It is particularly important to ensure that the description of the methods in the main text are as free from jargon as possible, and that the gist of what you did can be understood by general readers who are not experienced in the particular technical approaches you used. If necessary, technical details can be included in supplementary material.

If you are able to address the reviewers' concerns, the paper has the potential to make an important contribution to the literature.

Reviewer(s)' Comments to Author:
Referee: 1

Comments to the Author(s)

This concise, well-constructed study uses a powerful combination of experimental and theoretical approaches to study how the distribution of social information from multiple sources affects individual judgements. The topic is important and very timely given the ever-increasing importance of social media in influencing individual attitudes and behaviour, including in areas of vital importance for public health such as vaccine acceptance, mask-wearing, and so on. The statistical approach is sound. The paper is also extremely clearly written and presented, with excellent use of informative figures.

I would recommend only a small number of minor changes before the study is accepted for publication in Proceedings B.

Ecological validity: the experimental design is sophisticated and in several ways far better reflects social influence in natural conditions compared with similar predecessors. In particular, allowing participants to view the judgements of multiple ‘demonstrators’ and integrate social with personal information to form ‘compromise’ decisions is much more realistic than other designs common in the field such as ‘either/or’ tasks where participants must choose between individual judgements and social information (often from a single demonstrator). However, as for any experimental task, the authors should consider the ways in which the study does not well represent ‘real-world’ conditions. Here, the task of guessing the number of animals in a herd is quite far abstracted from the social and political contexts the authors wish to generalise their findings to. Most importantly, in this study’s online task the demonstrators are anonymous strangers with no social connection to the participants, which may partly help to explain the relatively minor influence of social information relative to individual decisions the authors find here. While many contemporary online social spaces do involve interactions between anonymous strangers, individuals typically still have a consistent identity over time along with signifiers of expertise and affiliation which affect their level of influence. For example, my opinions on the state of the UK’s legal system would probably be influenced far more by the “Secret Barrister” than by any completely random, anonymous stranger posting on Twitter. The authors should also consider potential effects of the presentation of the task over the findings. In particular, the presentation of social information as numeric quantities on a linear, sliding scale (ESM) has obvious advantages in terms of the clarity of the task to participants, but again does not well represent social influence in the real world, which often involves multiple, both qualitative and quantitative dimensions. For example, this task does not well represent the nature of social influence over somebody deciding which way to vote in an upcoming election. Of course, no experiment can perfectly represent real-world conditions, but the authors should give these issues some consideration when evaluating their findings.

Representativeness: the main text includes little information about the study sample. The information on recruitment and participant demographics should be moved from the ESM to the main text, or at least briefly summarised there. Additionally, the authors should include some further detail on the ethnic/cultural make-up of the sample and provide descriptive statistics, if possible. This is important because the unrepresentativeness of ‘WEIRD’ samples is by now well-established (Henrich et al. 2010) and the use of social information has been shown to vary between cultures (e.g. Mesoudi et al. 2014).

Deception: the author’s experimental design cleverly avoids the need for deception while maintaining control over the social information. This should be mentioned in the main text as well as the ESM as it is a strength of the study. The authors should, however, also consider the possibility that participants may not believe the social information is real, despite their best efforts, considering that deception is still quite common in online experiments.

Understanding checks: as presented (if I understand correctly!) it looks as if the correct answers for the understanding checks (ESM) are all selected by default. Is that right? If so, I’d suggest avoiding this in future as it allows participants to pass the check even if they are just “clicking through” without properly reading the questions.

Henrich, J., Heine, S. J., & Norenzayan, A. (2010). The weirdest people in the world?. *The Behavioral and brain sciences*, 33(2-3), 61–135.

Mesoudi, A. et al. (2014) ‘Higher frequency of social learning in China than in the West shows cultural variation in the dynamics of cultural evolution’, *Proc. R. Soc. B* 282: 20142209.

Referee: 2

Comments to the Author(s)

Here, I summarize my thoughts on the submission, "Strategies for integrating disparate social information" to Proc B by Molleman et al. The key idea behind this study is straightforward, compelling, and insightful. At the root of every cultural evolutionary process is a distribution over social learning strategies. For many years, researchers working in this field were modellers, and what we had was a profusion of models, none of them with an especially sound empirical basis. Luckily, for roughly 10 years now, we've had a growing tradition of controlled experiments specifically designed to identify the social learning strategies that drive cultural evolutionary dynamics. All in all, this new field is a welcome change, but it's also highly unsettled. There are many paradigms, with little to help us sort out the relative benefits and liabilities of the different paradigms. Into this mix, we have the present study, which takes a clever new approach to a fundamental problem. Basically, the decision-making task can be viewed as a task with a large but finite number of options. Specifically, subjects had to guess the number of animals presented in an image. The key innovation is this. After making an initial choice, subjects were presented with social information as a set of three choices by others. The subject could then revise. What's clever here is that the authors manipulated the distribution of choices among the three observed individuals. Fig 1e shows the basic manipulations. Intuitively, (a) three others, close to each other, (b) three others, far apart from each other. (c) three others, with one close to the subject's initial choice and two far away, and (d) three others, with two close to the subject's initial choice and the other far away. For all four treatments, the difference between the subject's initial choice and the average of the three observed choices was constant. The question then is, how do the properties of the distribution characterizing observed social information shape how people learn socially? This is an important question as it goes straight to core issues in the study of cultural evolution and gene-culture coevolution. Namely, how do we actually learn from others? In spite of what the modellers would say, we don't actually know much about this. In this study, the particular question is, how do we actually learn from others when we observe a distribution over a large option space (probably a common situation)? In addition, how do the ways in which we learn vary by individual and/or context? We really don't know much about this, although Mesoudi (2016) is a nice review, and the first author of this submission has also done some nice work on the topic.

The current study has considerable potential, and I encourage the authors to continue with the interesting and important line of research. But, the paper would require a number of careful revisions before being suitable for publication in Proc B. I detail below, with points more or less in the order they come up in the paper itself.

- Where's the biology? At present, the paper is written almost as if it's intended for a psychology journal, perhaps a sociology journal, or maybe a general science journal. In some ways, Proc B has the flavour of a general science journal. But finally, it's a biology journal. The biology is basically missing from the submission as currently formulated. Ordinarily, one might think that this would be devastating for the submission, but I don't think that's true here. It would be easy - even trivial - to motivate the present study with fundamental questions related to evolutionary ecology, behavioural ecology, and in particular the evolutionary study of human social behaviour. For some reason, the authors did not do so for their initial submission, but I'm sure they can, and I believe they should. I would recommend discussing, at minimum, social learning and cultural evolution, heterogeneity in social learning, and the ever-controversial hypothesized link between culture and population structure.

- The description of the experiment in the main paper is missing key details. In particular, I couldn't tell from the main paper if choices/accuracy were incentivised, and I couldn't see how the authors could possibly implement the key treatment manipulations (LN vs HN vs HF vs HC) without using deception. As it turns out, choices were incentivised, and the authors did not use deception. I had to dig through the supplement to find this information, though, and even then the means by which the authors avoided deception still wasn't entirely clear. Regardless, given

that incentivised choices and deception are two points of disagreement in contemporary experimental work, the authors should explain these details in the main paper instead of leaving the reader to wonder and speculate.

- On related notes about the experimental design, I could not tell from the main paper what kind of feedback participants had at the end of each round. From the supplement, I think, but I'm not entirely sure, that participants did not get feedback about how close their respective guesses in a given round were to the correct answer. If correct, this information should be included in the main paper. In effect, even if each round gets a new stimulus (i.e. image with some number of animals), feedback at the end of each round would still allow participants to learn which social learning strategies are relatively effective, which in turn could mean that social learning strategies evolve endogenously throughout the course of a session. If participants did not get this kind of feedback, then this kind of learning is not possible. Depending on one's objectives, neither design is clearly better than the other, but the reader needs to know which one was implemented.

Finally, even without deception, the design, though clever, raises the following potential concern. In 20 out of 30 rounds, participants were provided with social information that followed extremely regular patterns. In effect, social information in these 20 rounds exhibited four highly regular and highly stylized distributions. I didn't get a strong sense how this was experienced by participants. The concern is that, if the social information looks too regular, participants might question whether it's real even if it is. The point is that the authors simply need to put more description of the experiment in the main paper. I know space is tight, so they'll have to be disciplined. But, experimentally minded readers will want a clear sense of the paradigm from a participant's perspective.

- As two final notes on the experiment/data, everything has been uploaded to OSF. However, from looking around on the repository, it looks like the authors did not pre-register the study. Rather, they simply uploaded everything over the course of two days after they were done. Also, they ran the study on mTurk. If I've understood correctly, neither of these choices is entirely consistent with current methodological trends related to experimental studies on evolution and human behaviour. At least, the authors should explain their reasons.

- Fig. 2 requires more explanation to be interpretable.

- The explanation of the statistical methods was too terse and too ambiguous to understand. The supplement didn't help that much. If I had to trace my difficulties to something specific, I would say that the challenge follows from a tendency to use jargon as if it's standard terminology understood by all. From p. 10 onward, I see two different paths, either of which would probably be an improvement. The first possible path would involve considerably fewer technical details in the main paper. Just give the basic idea, refer to the supplement (which should definitely have a much more explicit, complete, and detailed exposition of the technicalities), and move on to the results.

The second path would involve more technical details in the main paper. To illustrate, what exactly do the authors mean by a "Bayesian mixture model"? What is the "standard logistic function"? Especially confusing, the use of the phrase "standard logistic function" implies a function that generates a probability distribution over a support with cardinality two. The authors, however, are modelling an outcome space with cardinality three ({keep, adopt, compromise}). This led me to ask, why are they using logistic when multinomial would be more appropriate? The paper outlines two models, with $P(\text{keep})$ and $P(\text{adopt})$ respectively on the left-hand sides. Each of these models is based on the standard logistic function. I naively take this to mean that they specify two models, one of which models a probability distribution over $\{P(\text{keep}), P(\text{not keep})\}$, and the other of which models a probability distribution over $\{P(\text{adopt}), P(\text{not adopt})\}$. If correct, how does one ensure that $P(\text{keep}) + P(\text{adopt}) \leq 1$? I couldn't really form an opinion, however, because the key details weren't presented.

Also, why are beliefs treated as discretised normal distributions? Why not simply use a distribution with non-negative integers as a support? Maybe this is something people do when they fit these kinds of models. I don't know. But the motivation for such an approach is not obvious. A normal distribution, of course, puts mass on negative numbers, which are obviously not possible. The need to discretise also suggests that normal is the wrong distribution anyway.

The issues mentioned here are representative of the basic challenge. The paper in its current form provides just enough technical detail to confuse, but not enough to ultimately sort it out and form an opinion in one's head. So, I would recommend one of the two paths outlined above.

Regardless of which path, the full details should be in the supplement to a much greater extent than what we have now.

- The bottom of p. 11 is not a sentence. I didn't understand the intended meaning.

- Finally, the simulations also require more explanation. Also, this section offers an opportunity to make stronger ties to biology. As currently framed, the discussion of the simulations reads like something targeted at social psychologists or even political scientists. While we're all no doubt fascinated and perhaps disgusted (alternatively delighted) by the social and political fragmentation destroying certain powerful countries, I'm not sure this should trump a discussion about fundamental biological questions in a submission to Proc B. Much of the discussion here, for example, could be linked to basic hypotheses about culture, population structure, and the potentially surprising evolutionary dynamics that might follow.

Author's Response to Decision Letter for (RSPB-2020-1310.R0)

See Appendix A.

RSPB-2020-2413.R0

Review form: Reviewer 2

Recommendation

Accept as is

Scientific importance: Is the manuscript an original and important contribution to its field?

Excellent

General interest: Is the paper of sufficient general interest?

Excellent

Quality of the paper: Is the overall quality of the paper suitable?

Excellent

Is the length of the paper justified?

Yes

Should the paper be seen by a specialist statistical reviewer?

No

Do you have any concerns about statistical analyses in this paper? If so, please specify them explicitly in your report.

No

It is a condition of publication that authors make their supporting data, code and materials available - either as supplementary material or hosted in an external repository. Please rate, if applicable, the supporting data on the following criteria.

Is it accessible?

Yes

Is it clear?

Yes

Is it adequate?

Yes

Do you have any ethical concerns with this paper?

No

Comments to the Author

The revision is really excellent. Thank you for your careful work, and I look forward to seeing the paper in print.

Decision letter (RSPB-2020-2413.R0)

29-Oct-2020

Dear Dr Molleman

I am pleased to inform you that your manuscript RSPB-2020-2413 entitled "Strategies for integrating disparate social information" has been accepted for publication in Proceedings B.

The referee(s) have recommended publication, but also suggest some minor revisions to your manuscript. Therefore, I invite you to respond to the referee(s)' comments and revise your manuscript. Because the schedule for publication is very tight, it is a condition of publication that you submit the revised version of your manuscript within 7 days. If you do not think you will be able to meet this date please let us know.

Sincerely,
Dr Robert Barton
mailto: proceedingsb@royalsociety.org

Associate Editor
Comments to Author:

Thank you for your thorough efforts in revising the manuscript. The reviewer and I both agree that the revisions have improved the paper, and that it will make a great contribution to the literature.

Reviewer(s)' Comments to Author:

Referee: 2

Comments to the Author(s).

The revision is really excellent. Thank you for your careful work, and I look forward to seeing the paper in print.

Decision letter (RSPB-2020-2413.R1)

30-Oct-2020

Dear Dr Molleman

I am pleased to inform you that your manuscript entitled "Strategies for integrating disparate social information" has been accepted for publication in Proceedings B.

Open Access

Paper charges

Sincerely,
Proceedings B
mailto: proceedingsb@royalsociety.org

Appendix A

Dear Dr. Barton,

Thank you for your invitation to resubmit a revised version of our manuscript "Strategies for integrating disparate social information" (RSPB-2020-1310). We thank you, the Editorial Board Member and both reviewers for the insightful and constructive comments.

We have revised our paper in light of the comments and made several changes. Below, we outline our responses to the comments (in blue), documenting how and where we have changed the manuscript.

We hope our revised version is acceptable for publication in *Proceedings of the Royal Society B*.

Yours sincerely,

Lucas Molleman

Editorial Board Member

The two reviewers agree, as do I, that this study takes a very clever approach to address interesting and important questions in cultural evolution. However, both reviewers raise a number of important concerns that need to be addressed in revision.

In particular, both reviewers highlight a lack of detail on the likely ecological relevance of the study (as reviewer 2 puts it, "where's the biology"). Emphasising how the study may reflect real-life problems and linking it more clearly to the literature on social learning, cultural evolution and population structure will help to address these concerns.

We thank the Board Member for his/her constructive comments. We agree that our paper stood to benefit from stronger connections to the literature on social learning and cultural evolution. We have rewritten our manuscript accordingly, linking it to both of these topics, most notably in the abstract (lines 21-26), introduction (lines 43-56), the simulation results (lines 307-335) and the discussion (lines 415-449).

Both reviewers also request further details and clarification on aspects of the experimental and analytical methods. It is particularly important to ensure that the description of the methods in the main text are as free from jargon as possible, and that the gist of what you did can be understood by general readers who are not experienced in the particular technical approaches you used. If necessary, technical details can be included in supplementary material.

We have rewritten the Experimental Design section in the main text to make these as accessible as possible for the broad readership of *Proceedings B*. A full description of the experimental methods and the model - including all technical details - is now given in the Electronic Supplementary Material (ESM).

If you are able to address the reviewers' concerns, the paper has the potential to make an important contribution to the literature.

Referee: 1

This concise, well-constructed study uses a powerful combination of experimental and theoretical approaches to study how the distribution of social information from multiple sources affects individual judgements. The topic is important and very timely given the ever-increasing importance of social media in influencing individual attitudes and behaviour, including in areas of vital importance for public health such as vaccine acceptance, mask-wearing, and so on. The statistical approach is sound. The paper is also extremely clearly written and presented, with excellent use of informative figures.

We thank the reviewer for his/her positive comments.

I would recommend only a small number of minor changes before the study is accepted for publication in Proceedings B.

Ecological validity: the experimental design is sophisticated and in several ways far better reflects social influence in natural conditions compared with similar predecessors. In particular, allowing participants to view the judgements of multiple 'demonstrators' and integrate social with personal information to form 'compromise' decisions is much more realistic than other designs common in the field such as 'either/or' tasks where participants must choose between individual judgements and social information (often from a single demonstrator). However, as for any experimental task, the authors should consider the ways in which the study does not well represent 'real-world' conditions. Here, the task of guessing the number of animals in a herd is quite far abstracted from the social and political contexts the authors wish to generalise their findings to. Most importantly, in this study's online task the demonstrators are anonymous strangers with no social connection to the participants, which may partly help to explain the relatively minor influence of social information relative to individual decisions the authors find here. While many contemporary online social spaces do involve interactions between anonymous strangers, individuals typically still have a consistent identity over time along with signifiers of expertise and affiliation which affect their level of influence. For example, my opinions on the state of the UK's legal system would probably be influenced far more by the "Secret Barrister" than by any completely random, anonymous stranger posting on Twitter. The authors should also consider potential effects of the presentation of the task over the findings. In particular, the presentation of social information as numeric quantities on a linear, sliding scale (ESM) has obvious advantages in terms of the clarity of the task to participants, but again does not well represent social influence in the real world, which often involves multiple, both qualitative and quantitative dimensions. For example, this task does not well represent the nature of social influence over somebody deciding which way to vote in an upcoming election. Of course, no experiment can perfectly represent real-world conditions, but the authors should give these issues some consideration when evaluating their findings.

We agree that, depending on the exact context, there can be important differences between real-world scenarios of social information use and our experimental design. We have revised our

penultimate discussion paragraph (lines 424-449) highlighting the above considerations, providing a more balanced view of the limitations and generalisability of our study and results. At the same time, this allowed us to better connect to the literature on social learning and cultural evolution.

Representativeness: the main text includes little information about the study sample. The information on recruitment and participant demographics should be moved from the ESM to the main text, or at least briefly summarised there. Additionally, the authors should include some further detail on the ethnic/cultural make-up of the sample and provide descriptive statistics, if possible. This is important because the unrepresentativeness of 'WEIRD' samples is by now well-established (Henrich et al. 2010) and the use of social information has been shown to vary between cultures (e.g. Mesoudi et al. 2014).

We now summarize the participants' demographics (i.e., country of origin, age, and sex) in the Experimental Design section (lines 150-151). In addition, in our revised Discussion, we now qualify the generalizability of our results to other populations and cite the above-mentioned studies and other relevant literature (lines 425-437).

Deception: the author's experimental design cleverly avoids the need for deception while maintaining control over the social information. This should be mentioned in the main text as well as the ESM as it is a strength of the study. The authors should, however, also consider the possibility that participants may not believe the social information is real, despite their best efforts, considering that deception is still quite common in online experiments.

We now highlight the 'experimental control without deception' aspect of our study in the section 'Experimental Design' (lines 141-148), and provide full details in the ESM, section 3a.

We share the concern that some participants might not believe the information shown to them is actual social information generated by others in the experiment. We added the following brief note to the ESM, section 3a, header *Participants' belief that social information was real*:

"Estimates in our experiment were incentivised so that participants should only adjust them based on social information in case they think it will improve their accuracy. The fact that overall, adjustments were quite substantial, suggests that participants trusted the social information. Furthermore, in a separate study using a basic version of the BEAST paradigm (manuscript in preparation), we asked participants to elaborate on how they integrated the estimates of other MTurkers into their own judgments. Out of 209 participants, only three expressed doubt about the veracity of the social information shown to them (and hence chose to ignore it). This suggests that the vast majority of MTurkers tend to trust that the social information in our experiment was real."

Understanding checks: as presented (if I understand correctly!) it looks as if the correct answers for the understanding checks (ESM) are all selected by default. Is that right? If so, I'd suggest avoiding this in future as it allows participants to pass the check even if they are just "clicking through" without properly reading the questions.

We thank the reviewer for pointing this out. The correct answers to the control questions were NOT pre-selected, but we understand how the original screenshot may have suggested this. We have replaced the screenshot in the ESM (with the new one taken before the questions were answered).

Henrich, J., Heine, S. J., & Norenzayan, A. (2010). The weirdest people in the world?. *The Behavioral and brain sciences*, 33(2-3), 61–135.

Mesoudi, A. et al. (2014) 'Higher frequency of social learning in China than in the West shows cultural variation in the dynamics of cultural evolution', *Proc. R. Soc. B* 282: 20142209.

Referee: 2

Here, I summarize my thoughts on the submission, "Strategies for integrating disparate social information" to Proc B by Molleman et al. The key idea behind this study is straightforward, compelling, and insightful. At the root of every cultural evolutionary process is a distribution over social learning strategies. For many years, researchers working in this field were modellers, and what we had was a profusion of models, none of them with an especially sound empirical basis. Luckily, for roughly 10 years now, we've had a growing tradition of controlled experiments specifically designed to identify the social learning strategies that drive cultural evolutionary dynamics. All in all, this new field is a welcome change, but it's also highly unsettled. There are many paradigms, with little to help us sort out the relative benefits and liabilities of the different paradigms. Into this mix, we have the present study, which takes a clever new approach to a fundamental problem. Basically, the decision-making task can be viewed as a task with a large but finite number of options. Specifically, subjects had to guess the number of animals presented in an image. The key innovation is this. After making an initial choice, subjects were presented with social information as a set of three choices by others. The subject could then revise. What's clever here is that the authors manipulated the distribution of choices among the three observed individuals. Fig 1e shows the basic manipulations. Intuitively, (a) three others, close to each other, (b) three others, far apart from each other. (c) three others, with one close to the subject's initial choice and two far away, and (d) three others, with two close to the subject's initial choice and the other far away. For all four treatments, the difference between the subject's initial choice and the average of the three observed choices was constant. The question then is, how do the properties of the distribution characterizing observed social information shape how people learn socially? This is an important question as it goes straight to core issues in the study of cultural evolution and gene-culture coevolution. Namely, how do we actually learn from others? In spite of what the modellers would say, we don't actually know much about this. In this study, the particular question is, how do we actually learn from others when we observe a distribution over a large option space (probably a common situation)? In addition, how do the ways in which we learn vary by individual and/or context? We really don't know much about this, although Mesoudi (2016) is a nice review, and the first author of this submission has also done some nice work on the topic.

The current study has considerable potential, and I encourage the authors to continue with the interesting and important line of research. But, the paper would require a number of careful revisions before being suitable for publication in Proc B. I detail below, with points more or less in the order they come up in the paper itself.

- Where's the biology? At present, the paper is written almost as if it's intended for a psychology journal, perhaps a sociology journal, or maybe a general science journal. In some ways, Proc B has the flavour of a general science journal. But finally, it's a biology journal. The biology is basically missing from the submission as currently formulated. Ordinarily, one might think that this would be devastating for the submission, but I don't think that's true here. It would be easy - even trivial - to motivate the present study with fundamental questions related to evolutionary ecology, behavioural ecology, and in particular the evolutionary study of human social behaviour. For some reason, the authors did not do so for their initial submission, but I'm sure they can, and I believe they should. I would recommend discussing, at minimum, social learning and cultural evolution, heterogeneity in social learning, and the ever-controversial hypothesized link between culture and population structure.

We thank the reviewer for his/her constructive comments. We agree that for the broad (but biology-centred) readership of Proceedings B, our work requires solid embedding in these important biological topics. We rewrote our introduction to accommodate these points, locating our work better in the related literature on social learning and cultural evolution in human and non-human animals, and linking individual-level strategies and population structure to population-level consequences. In addition, we rewrote parts of the discussion, most notably, the penultimate paragraph where we reflect on the limitations and possible extensions of our study, again, improving the connection between our work and the social learning literature. We feel that these changes have substantially improved the paper and its potential appeal to the readers of this journal.

- The description of the experiment in the main paper is missing key details. In particular, I couldn't tell from the main paper if choices/accuracy were incentivised, and I couldn't see how the authors could possibly implement the key treatment manipulations (LN vs HN vs HF vs HC) without using deception. As it turns out, choices were incentivised, and the authors did not use deception. I had to dig through the supplement to find this information, though, and even then the means by which the authors avoided deception still wasn't entirely clear. Regardless, given that incentivised choices and deception are two points of disagreement in contemporary experimental work, the authors should explain these details in the main paper instead of leaving the reader to wonder and speculate.

We thank the reviewer for flagging this and we agree that this is important information. In response, we added these details to the Experimental Design section of the main text (lines 141-161). The relevant section now reads:

“Prior to the main experiment, we pre-recorded individual estimates for each of the images by 100 individuals recruited from Amazon Mechanical Turk (MTurk), rewarding them for accuracy (ESM, Section 3a). We used these estimates as social information in the main experiment. In a given round of the main experiment, the three pieces of pre-recorded social information were selected based on the participant's first estimate and the experimental condition of that round. That is, we selected those pieces of social information that most closely matched the experimental condition. This procedure allowed us to achieve experimental control without using deception (for full details and screenshots, see electronic supplementary materials (ESM), sections 3a and 4).

Ninety-five participants (all from the United States; 57% male; mean \pm s.d. age: 35.8 \pm 10.7 years) were recruited from MTurk for the main experiment, and completed 30 rounds of the judgment task. These 30 rounds included five rounds of each condition and 10 'filler' rounds. The social information in the filler rounds consisted of three randomly drawn estimates of the pre-recorded participants (for a given image). This procedure ensured that across all rounds, social information was sometimes higher and sometimes lower than a participant's first estimate, and sometimes bracketed the participant's own estimate. Reducing the regularity of the presented social information was expected to increase its trustworthiness. The 30 rounds were shown in a random order (and this order was the same for all participants). Throughout the task, participants did not receive feedback about their accuracy, impeding opportunities to learn about their own performance or the quality of the social information. Participants were rewarded for the accuracy: at the end of the experiment, one (first or second) estimate from the 30 rounds was selected and used for payment (see ESM, Section 3a for details)."

- On related notes about the experimental design, I could not tell from the main paper what kind of feedback participants had at the end of each round. From the supplement, I think, but I'm not entirely sure, that participants did not get feedback about how close their respective guesses in a given round were to the correct answer. If correct, this information should be included in the main paper. In effect, even if each round gets a new stimulus (i.e. image with some number of animals), feedback at the end of each round would still allow participants to learn which social learning strategies are relatively effective, which in turn could mean that social learning strategies evolve endogenously throughout the course of a session. If participants did not get this kind of feedback, then this kind of learning is not possible. Depending on one's objectives, neither design is clearly better than the other, but the reader needs to know which one was implemented. Finally, even without deception, the design, though clever, raises the following potential concern. In 20 out of 30 rounds, participants were provided with social information that followed extremely regular patterns. In effect, social information in these 20 rounds exhibited four highly regular and highly stylized distributions. I didn't get a strong sense how this was experienced by participants. The concern is that, if the social information looks too regular, participants might question whether it's real even if it is. The point is that the authors simply need to put more description of the experiment in the main paper. I know space is tight, so they'll have to be disciplined. But, experimentally minded readers will want a clear sense of the paradigm from a participant's perspective.

Regarding feedback: We agree this is important information. During the task, participants did not receive any feedback to reduce (performance-based) updating of learning strategies. We have added this information - including the motivation for this design choice - to the Experimental Design section (lines 158-161; relevant information is quoted above).

Regarding regularity of social information: As noted in the main text, participants completed 30 rounds consisting of five rounds of each experimental condition and 10 'filler rounds'. These were shown in random order to reduce the risk of detecting any regularity in the social information (lines 156-158). Also, the filler rounds, containing three randomly drawn pieces of social information were included with this aim. Moreover, a round consisted of the following separate screens: (i) stimulus, (ii) enter first estimate, and (iii) enter second estimate, showing the social information (Fig. 1a-c). That is, screens containing social information were not shown right after each other. Therefore, it

seems rather unlikely that participants identified a regular pattern across rounds. To give readers a sense of what participants experienced, we added an illustration of the 30 decision situations with social information to the ESM, right below the screenshots of the decision situation with social information (page 37).

To test whether participants changed their use of social information over time, potentially indicating a reduced trust in that information, we used a linear mixed model (LMM). This LMM was similar to the one reported in Table S1, but now fitted to adjustments in individual rounds (rather than participants' averages in each condition), and included 'round number' as an additional predictor to test for time trends in adjustments. If participants would lose faith in the veracity of social information, we would expect a downward trend in social information use over time. This idea is not supported by the model: we observe that 'period' does not reliably predict adjustments, suggesting that over time, individuals did not shift their social information use downwards. We report these results in the newly added Table S7 (also printed below, and referred to in the caption of Table S1).

	Estimate	95% CI
round	0.00	[0.00, 0.00]
LN	0.36	[0.18, 0.55]
HN	-0.13	[-0.16, -0.10]
HF	-0.14	[-0.16, -0.11]
HC	-0.05	[-0.08, -0.02]
age	0.00	[0.00, 0.00]
gender	0.04	[-0.06, 0.13]
N	95	
n	1839	

Results of a LMM fitted to adjustments in individual rounds. Treatment abbreviations are as in Table S1. 'Round' did not impact adjustments. Treatment effects are very similar to the ones reported in Table S1.

- As two final notes on the experiment/data, everything has been uploaded to OSF. However, from looking around on the repository, it looks like the authors did not pre-register the study. Rather, they simply uploaded everything over the course of two days after they were done. Also, they ran the study on mTurk. If I've understood correctly, neither of these choices is entirely consistent with current methodological trends related to experimental studies on evolution and human behaviour. At least, the authors should explain their reasons.

We indeed did not pre-register our study, and the reviewer rightly calls us out on this. Since 2018 (when we collected these data) we have implemented pre-registration as our standard practise, but for this manuscript, all we can do with regard to openness is share our code and data.

We chose to run our experiments on MTurk partly for convenience, but also because in previous studies, samples from MTurk have produced reliable and replicable data with our experimental paradigm (see Molleman, Kurvers and Van den Bos 2019 *EHB* 'Unleashing the BEAST - a brief measure for human social information use' for an extensive validation exercise). In addition, with this paradigm we have done multiple studies with participants in more traditional lab conditions

with university students, yielding results that are very similar to the ones obtained from MTurk. In response to this comment, we have added remarks on our task's validity on MTurk (ESM, section 3a, paragraph 2). We there state:

“The experimental task is based on a validated perceptual judgment paradigm for quantifying social information use (BEAST [ref. 1]; Fig. 1a-c). The basic version of this task has been used in samples from various ages and cultural backgrounds [refs. 1,7,8], and has been shown to have high test-retest reliability with participants recruited from MTurk [ref. 1]. ”

Furthermore, in the penultimate paragraph of the Discussion in the main text (lines 425-437), we note that the generalizability of our current results to other samples remains an empirical question.

- Fig. 2 requires more explanation to be interpretable.

In our revision we have added more information to this figure caption, taking the reader more by the hand in explaining the observed patterns.

- The explanation of the statistical methods was too terse and too ambiguous to understand. The supplement didn't help that much. If I had to trace my difficulties to something specific, I would say that the challenge follows from a tendency to use jargon as if it's standard terminology understood by all. From p. 10 onward, I see two different paths, either of which would probably be an improvement. The first possible path would involve considerably fewer technical details in the main paper. Just give the basic idea, refer to the supplement (which should definitely have a much more explicit, complete, and detailed exposition of the technicalities), and move on to the results.

The second path would involve more technical details in the main paper. To illustrate, what exactly do the authors mean by a "Bayesian mixture model"? What is the "standard logistic function"? Especially confusing, the use of the phrase "standard logistic function" implies a function that generates a probability distribution over a support with cardinality two. The authors, however, are modelling an outcome space with cardinality three ($\{\text{keep}, \text{adopt}, \text{compromise}\}$). This led me to ask, why are they using logistic when multinomial would be more appropriate? The paper outlines two models, with $P(\text{keep})$ and $P(\text{adopt})$ respectively on the left-hand sides. Each of these models is based on the standard logistic function. I naively take this to mean that they specify two models, one of which models a probability distribution over $\{P(\text{keep}), P(\text{not keep})\}$, and the other of which models a probability distribution over $\{P(\text{adopt}), P(\text{not adopt})\}$. If correct, how does one ensure that $P(\text{keep}) + P(\text{adopt}) \leq 1$? I couldn't really form an opinion, however, because the key details weren't presented.

We thank the reviewer for this comment, and for sketching clear paths for improving our manuscript. We implemented path 1, presenting the basic conceptual idea behind our model in the main text, and moving its details into a dedicated section in the ESM (section 3b). We extended the model descriptions and motivations behind modeling choices (see also below). Doing so, we provide all model details in full while trying to avoid jargon (e.g., removing the unnecessary references to the term 'Bayesian mixture model', explicitly specifying the logistic functions right where they appear,

and indicating how the logistic functions make sure that $P(\text{keep}) + P(\text{adopt}) + P(\text{compromise}) = 1$. The reviewers questions helped us to present the model in a much clearer way.

Also, why are beliefs treated as discretised normal distributions? Why not simply use a distribution with non-negative integers as a support? Maybe this is something people do when they fit these kinds of models. I don't know. But the motivation for such an approach is not obvious. A normal distribution, of course, puts mass on negative numbers, which are obviously not possible. The need to discretise also suggests that normal is the wrong distribution anyway.

We thank the reviewer for bringing this up and we agree this section was unclear. There are two related issues here (i) why use a normal distribution? and (ii) why discretize these distributions? In the ESM, Section 3b, under the header "Compromising modelled as a process of Bayesian updating." we explicitly motivate our modeling choices. We added:

"For modelling the quantity judgments in our task, normal distributions are a natural choice to represent uncertainty around a point estimate. These distributions have two desirable properties. First, their probability density is highest at the centre. This seems reasonable as in our case, participants are incentivised to enter values they deem most likely. Second, the probability density is symmetrically decreasing as values are further away from the centre. Indeed, using normal distributions to represent uncertainty round a point estimate is a common approach in Bayesian models of belief updating, including models considering social information [ref. 18].

These distributions are typically modelled as continuous density functions. Note that in our experiments, values of 'beliefs' were restricted to integer numbers from 1 to 150 (the range of the slider for entering estimates). To reflect this, and to make our analysis of compromising consistent with heuristics of keeping and adopting (by definition also restricted to integers), we discretised the normal distributions by calculating the relative probability of each integer from 1 to 150 and normalizing the sum of all probabilities to 1."

The issues mentioned here are representative of the basic challenge. The paper in its current form provides just enough technical detail to confuse, but not enough to ultimately sort it out and form an opinion in one's head. So, I would recommend one of the two paths outlined above. Regardless of which path, the full details should be in the supplement to a much greater extent than what we have now.

As mentioned above, in the main text of the revision we now focus on the conceptual aspects of our model, and present all technical details in one dedicated section in the ESM (section 3b).

- The bottom of p. 11 is not a sentence. I didn't understand the intended meaning.

We have rewritten the sentence "Note that compromising results in a probability distribution for adjustments, which might result in instances of keeping or adopting." as "Note that compromising generates a probability distribution of updated (second) estimates, which might include the individuals' own first estimate, and the estimate of a peer. As a consequence, compromising might

result in instances of keeping or adopting.” Please note that this sentence is now part of the ESM (Section 3b), following our reply to the previous comment.

- Finally, the simulations also require more explanation. Also, this section offers an opportunity to make stronger ties to biology. As currently framed, the discussion of the simulations reads like something targeted at social psychologists or even political scientists. While we're all no doubt fascinated and perhaps disgusted (alternatively delighted) by the social and political fragmentation destroying certain powerful countries, I'm not sure this should trump a discussion about fundamental biological questions in a submission to Proc B. Much of the discussion here, for example, could be linked to basic hypotheses about culture, population structure, and the potentially surprising evolutionary dynamics that might follow.

When describing the simulations, we now explicitly link these from the outset to the (biological) themes from the introduction, highlighting that the studied settings correspond to “individuals’ access to information from their social network being local or global [ref. 45], their personal preferences for homophily [ref. 46], or, in case of human online interactions, being controlled by algorithms prioritising similar (or dissimilar) social sources over others, biasing the available social information [ref. 47].” (lines 309-312).

Throughout describing our results, and discussing them, our revised manuscript focuses less on the specifics of polarization, but makes explicit links to social network structure more broadly. We discuss our results in relation to how population structure may interact with (individual differences in) social information use to shape dynamics of cultural evolution (lines 415-422).